# Overcoming Alzheimer’s Disease Stigma by Leveraging Artificial Intelligence and Blockchain Technologies

**DOI:** 10.3390/brainsci10030183

**Published:** 2020-03-23

**Authors:** Alexander Pilozzi, Xudong Huang

**Affiliations:** Department of Psychiatry, Massachusetts General Hospital and Harvard Medical School, Charlestown, MA 02129, USA; apilozzi@mgh.harvard.edu

**Keywords:** Alzheimer’s disease, artificial intelligence, natural language processing, sentiment analysis, social media, blockchain, stigma

## Abstract

Alzheimer’s disease (AD) imposes a considerable burden on those diagnosed. Faced with a neurodegenerative decline for which there is no effective cure or prevention method, sufferers of the disease are subject to judgement, both self-imposed and otherwise, that can have a great deal of effect on their lives. The burden of this stigma is more than just psychological, as reluctance to face an AD diagnosis can lead people to avoid early diagnosis, treatment, and research opportunities that may be beneficial to them, and that may help progress towards fighting AD and its progression. In this review, we discuss how recent advents in information technology may be employed to help fight this stigma. Using artificial intelligence (AI) technologies, specifically natural language processing (NLP), to classify the sentiment and tone of texts, such as those of online posts on various social media sites, has proven to be an effective tool for assessing the opinions of the general public on certain topics. These tools can be used to analyze the public stigma surrounding AD. Additionally, there is much concern among individuals that an AD diagnosis, or evidence of pre-clinical AD such as a biomarker or imaging test results, may wind up unintentionally disclosed to an entity that may discriminate against them. The lackluster security record of many medical institutions justifies this fear to an extent. Adopting more secure and decentralized methods of data transfer and storage, and giving patients enhanced ability to control their own data, such as a blockchain-based method, may help to alleviate some of these fears.

## 1. Introduction

Dementia is unique in terms of the societal challenges it poses, and Alzheimer’s disease (AD) is the most common cause of dementia [1]. The disease affects around 10% of the population 65 years and over, and nearly a third of those over the age of 85. Further, it is estimated that 5.8 million Americans currently live with the disease, a number that is projected to continue to rise along with an aging population. Unfortunately, there is currently no effective treatment or prevention for the disease [1]. Along with the impairments that come with the disease, AD sufferers also experience internalized and public stigma, as they face judgement from themselves and those around them for their disease and their symptoms [2]. Disease prognosis bears heavily on stigma development; stigma surrounding has much to do with the expectation of cognitive decline associated with the condition, rather than the name itself [3]. The diagnosis of AD carries an association between the person diagnosed and the symptoms that come along with it, often regardless of whether those symptoms are present or not [2]. These perceptions can negatively impact a patient’s ability to experience a happy and meaningful life in the time between diagnosis and the development of more severe symptoms [4]. A 2012 survey found that over 75% of those living with dementia believe themselves to be subjected to negative associations, with around a quarter believing they were marginalized by others (28%) or that others did not know how to interact with them (24%) [5]. Furthermore, many people believe that institutionalized stigma, in the form of employment and insurance discrimination, among others, would beset a person diagnosed with AD [6]. The burden of this stigma is more than just psychological; the prevalence of AD stigma poses a significant barrier to early diagnosis and treatment of the disease, while also discouraging individuals from participating in clinical trials [7].

## 2. Utilizing Natural Language Processing to Fight Stigma

### 2.1. Social Media as a Propagator of and Protector from Stigma

The widespread adoption of social media has its benefits for many patients. Patients are now able to connect with others, and share their experiences in health-care or with a specific illness [8]. Networks of users with common ailments have formed, where they can share information, resources, and support amongst each other [9]. The organization of a group, as well as who it includes, is integral to defining the group’s effects. Groups can be beneficial in helping individuals in fighting stigma and reduce its internalization [10,11]. However, the maintenance of ingroups and outgroups in communities of those with stigmatizing experiences can have an isolating and polarizing effect on its members, with them avoiding stigma rather than fighting it, ultimately having negative consequences for the group participants [12,13]. As much of the research into the impact of these groups has been done on those suffering from stigmatized mental illness, it is unclear how these findings translate to AD. AD and other dementias are unique in their situation as physical, neurodegenerative ailments with cognitive effects, featuring aspects of both physical debilitation and mental illness stigma that blend together [2].

Unfortunately, social media also provides platforms for discriminatory speech that may contribute to stigma. A 2013 study looking at how Facebook users describe older (60+) individuals found that out of 84 Facebook groups with descriptions pertaining to those over 60 years old found that 74% excoriated them, 27% infantilized them, and 37% wanted them banned from certain public activities; all but one were disparaging to the elderly in some way. Of these descriptions 27% referred to some cognitive debilitation (dementia, “senile”, etc.), and 13% referred to both cognitive and physical debilitations [14]. The psychological harm incurred by these stereotypes does not only extend to the minds and attitudes of older individuals, but also to their physical well-being. Older individuals who expressed more positive stereotypes about the elderly were found to fully recover from severe daily-life-inhibiting disabilities 44% more often than those who held negative beliefs/stereotypes [15].

While stigma is a prevalent problem for a wide variety of different health and social issues, sources of information from which to develop an understanding of how it develops and propagates are difficult to come by. The advent of social media and other electronic communications has opened up new avenues for sentiment research and opinion mining, among other applications. It has given rise to a relatively new field known as “Infodemiology”, a term coined by Gunther Eysenbach, which is defined as “the science of distribution and determinants of information in an electronic medium or in a population with the ultimate aim to inform public health and public policy” [16]. Though largely concerned with how users post, consume, or search for information online, it can be used to predict and analyze trends in health and healthcare as well as identify potential disease outbreaks. The methodology used in processing free-text posts and extracting health-relevant information can be applied to discerning the public perception of certain health-related topics [17]. Given social media’s presence as a platform from which individuals can broadcast their opinions, it comes as no surprise that considerable effort is devoted to analyzing trends in how people generally feel about certain entities [18].

### 2.2. Natural Language Processing in Stigma Research

The overarching goal of the field of natural language processing (NLP) is to analyze unstructured text, and generate a structured representation of that text [19]. This typically involves digitizing a set of documents or importing already-digital communications and applying some rule-based or machine learning classification system to process, analyze, and/or categorize various textual entities [20]. Such systems have been utilized in the medical field primarily for the extraction of certain information from various medical documents such as pathology reports [19]. Computational NLP tactics have become increasingly popular over the years, with an average growth rate for NLP-based medical research publications of approximately 18.39% between 2007 and 2016 [21]. With regard to stigma and sentiment, however, mining medical data for relevant information is difficult as there is little sentiment to be gleaned from documents aiming to be as objective as possible [22]. That is where the wealth of information contained in social media posts comes into play.

### 2.3. Sentiment in Social Media

Manual annotation of social media posts by multiple reviewers has been used to analyze sentiment in social media. Based on manual classification, around 31% of Facebook and Twitter posts related to cancer and obesity exhibited some form of sentiment [23]. Manual annotation is a difficult, and sometimes inconsistent, process [24]; thus the development of automated sentiment analysis solutions is imperative to the field as a whole. However, while there are a wide array of open source and commercially licensed sentiment analysis pipelines, most are targeted toward consumer-based/marketing fields, with limited resources verified for applicability to health/health-care sentiment [24]. With that being said, there have certainly been some meaningful applications of both pre-developed tools and methods developed by the researchers themselves. Using a term-based support vector machine (SVM), researchers were able to classify the overall sentiment of twitter status posts regarding smoking with 85.6% accuracy relative to manual annotation [25]. Analysis of Facebook and Twitter posts related to obesity and cancer found that 31% had some sort of sentiment; it was found that negative individual-blaming sentiments dominated obesity related postings, while there was greater support and empathy for survivors of cancer [23]. On a similar note, tweets involving type II diabetes, a disease that is conventionally thought of as lifestyle-related, were considerably more negative than tweets involving type 1 diabetes [26]. These sentiment-classified messages can be further mined for more information on the nature of the sentiment. Patterns in the wording of negative-sentiment messages can elucidate to whom these messages are targeted, and what negative terminology is associated with the subject in question. A 2014 study found that many derogatory messages associated with obesity exhibited a significant focus on individual blame, with a notable trend in misogynistic undertones [27]. These online messages certainly contribute to the propagation of stigma, and the methods used to analyze them are applicable to related media.

The ability to measure overall negative sentiment/stigma has benefits for efforts in quashing that stigma. As of now, the vast majority of anti-stigma campaigns have no measurement of results; as such it is unclear whether these efforts are effective and worthwhile, if resources should be directed elsewhere, or if a change in tactics is in order [28]. Measuring stigma before and after a campaign is introduced would allow for a meaningful analysis of the result, provided sufficient data is taken and/or regular monitoring of sentiment regarding relevant keywords takes place.

### 2.4. NLP in Chat-Bots/AI Companions

A considerable problem facing AD patients and older individuals more generally is social isolation. It is not only a facet of the stigma they face [29], but also an apparent detriment to their cognitive health. Indeed, social isolation tends to increase with age [30,31], and the level of isolation has been found to be associated with the profundity of memory decline [31,32], and increases in various signs/symptoms of cognitive decline [33]. More generally, loneliness in old age is a significant risk factor for poor health outcomes, death, and functional decline [34]. Given the physical and mental health impact of social isolation and loneliness methods to combat it are vital; especially so considering 90% of adults over 65 years old plan to/want to stay in their homes as long as they can, leaving them vulnerable to isolation [35].

Methods of both ameliorating this social isolation, and simultaneously providing utility to AD patients, and older individuals more generally, may come from the use of AI “chat-bots”, which utilize NLP to understand and respond to user queries. The allure of such systems in providing services to the elderly is understandable. The global population is aging considerably, while there is a proportional decrease in social and healthcare providers [36]. Automation of some caretaking tasks, such as general social interaction, may be integral to caring for an increasing number of dementia patients.

The implementation of AI for social/companionship roles seems to be useful. Simple animal-like robots have been found to reduce the mean loneliness scores in older individuals [36,37,38]. Socially assistive robots (SAR) designed for social facilitation have been found to increase social interactions with the subject’s peers as well [36]. The SAR Paro, a robotic baby harp-seal designed to be expressive as well as capable of responding to users’ voices and actions [39], is well studied, particularly with regard to dementia patients. Researchers find that Paro generally improves mood, reduces loneliness and alleviates stress in dementia patients [37,40]. However, perhaps owing to the lack of complex social behavior, some studies find Paro’s ability to lower aggression and reduce neutral affect to be comparable to a stuffed toy, calling into question the cost effectiveness of robot-therapy [41,42].

The creation of virtual agents that are capable of conversing with users in real time is challenging but may be worthwhile. Older individuals expressed significant interest in communicating regularly with a “Wizard of Oz” (human-controlled avatar/agent) agent on their computers. Users remarked on the connection they felt with the agent, despite knowing it was computer generated. However, while most users were satisfied and would like to continue working with the agent, some complained of its simplicity and lack of realism, ultimately failing to develop a connection [43]. Designing such an autonomous agent for use by the elderly and AD patients has its own unique set of challenges that need to be addressed; given their proclivity to make erroneous commands, proper error-recovery protocols are necessary to ensure smooth, natural communication [44].

It should also be noted that, presently, there is some apprehension in the older population towards the idea of a “robot”. Though practical data suggests that they are helpful, a focus group of individuals living with mild cognitive impairment (MCI) disagreed with the idea that a robot could serve as a companion, given a lack of real emotion/attachment [45]. Some of this opposition seemed to relate to preconceptions about robotics, such as an assistive robot being large, clumsy, and capable of causing damage to its surroundings [45]. However, individuals who are more familiar with robotics and its potentials tend to be more optimistic about the capabilities of a social robot [46]. Furthermore, tests with more advanced, humanoid SARs capable of conversation and emotional expression, such as the “Ryan companion bot” and “Brian 2.1”, are generally well received by users [47,48]. Perhaps demonstrating the capabilities of social robots to skeptical patients could alleviate their apprehensions. Interestingly, the presence of a physical body for an AI companion is a great boon to the development of an attachment [49,50]. Unfortunately, a motile humanoid chassis can be quite expensive [51], though cutting out mobility and other features can reduce the cost considerably [52]. Regardless, verbal input and output are highly prioritized by the potential elderly users of SARs [53,54], and a robust NLP platform is required for both processing and responding to user requests and conversation. Given the promising results of research involving the introduction of robots and virtual companions, it seems a worthwhile endeavor despite apparent opposition.

### 2.5. Natural Language Processing in AD Stigma Analysis & Research Direction

While applying NLP to online postings in order to analyze opinions surrounding a subject has considerable precedent, there have been few efforts made to apply the technique to the nature and development of related stigma. A study was conducted in 2017 to analyze the sentiment of over 31,000 Alzheimer’s-related tweets collected over a 10-day period, while also rating them on other contextual dimensions of the posts such as whether the post was a joke, meant to be informative, etc. It was found that 21.13% of these tweets were classified as negative/ridicule, with the majority (55%) of those comments being made in a joking context [55]. No other studies were readily available.

We propose that more efforts should be made to analyze the online presence of stigma. Having data from more sample periods is integral to understanding how prevalent AD stigma is online. Indeed, before measures can be taken to reduce the stigma of diseases such as Alzheimer’s, it is important to have established data on where the stigma exists and in what quantities both before and after campaigns are put in place in order to determine if the applied tactics are working effectively. Recent works have indicated that deep learning methods may outperform the standard supervised-learning models [56]. It may be fruitful to employ such a model in the study of AD stigma, ideally yielding more accurate classifications and a replicable method for analyzing stigma from online posts that can be employed at multiple time periods.

Furthermore, there are notable differences in the perception of dementia and aging between cultures [57]. However, the study performed on AD-related tweets, and indeed most social media NLP studies to date, have focused solely on English-language tweets and online postings. It may be worthwhile to examine posts in different languages to see if they exhibit the same sentiment proportions to get a better idea of the cultural differences in AD stigma.

As with many issues both medical and social, the key to resolving Stigma lies in identifying where its sources are. The advent of social media has generated hotbeds of opinions on a wide variety of subjects, including diseases such as Alzheimer’s. Some of these can be beneficial to disease-sufferers, who may find support amongst others like them or those who otherwise empathize with them; however, there are also some who disparage those with ailments such as AD and other dementias, which contributes to the perpetuation of stigma. In the current era it is imperative that the impact of social media on stigma be analyzed and addressed, and systems such as NLP sentiment-analysis are integral to making this possible. 

Utilizing NLP techniques in tandem with AI development in order to create a reasonably lively AI assistant that can provide both assistance and companionship would be a great boon to Alzheimer’s patients. The decreased feelings of social isolation, and potential for social facilitation, would allow for a greater degree of independence for AD patients. A summary of our proposed NLP solutions can be found in Figure 1.

## 3. Utilizing Blockchain Technologies to Fight Stigma

### 3.1. Information Privacy and Stigma

While in an ideal world people would feel comfortable and able to learn of their AD status and freely discuss it with others, that is not necessarily the case. Information privacy is a considerable concern in today’s world. There is the perception that an unintended leak about one’s AD status can have a great deal of impact on their life. Results of a survey published in 2018 found that the majority of their respondents believed that one diagnosed with the disease would be faced with employment discrimination. Additionally, almost half believed that Insurance would be limited based on an AD individual’s medical records or brain imaging results [6].

Indeed, these fears are certainly not unwarranted. While protections exist to mitigate genetic discrimination by employers and insurance companies, there are limited comparable protections for biomarker tests or imaging in the United States [58]. A person who partakes in AD related research, which may require the presence of certain biomarkers and features to qualify, is putting themselves at risk of facing discrimination prior to the onset of any cognitive impairment, as study data is often incorporated into the participant’s medical record [58].

A 2017 review found that that healthcare lags behind other fields in terms of security, calling into question the security of patient’s private health information [59]. From the beginning of 2019 to October there were over 38 million breaches in health records in the United States, with the largest individual breaches affecting over 100,000 people. There were over 500,000 records breached in hacking incidents in October alone. [60]. Concerns over the security of one’s medical record are real. The 2011–2012 HINTS survey found that 64.5% of respondents were concerned about a breach in the privacy and security of electronically transferred private health information. A further 12.3% reported having withheld information from healthcare professionals due to the fear of inadequate privacy and security [61]. The ineffective security practices surrounding EHR should be taken seriously, as patient confidence is integral to medical and research efforts.

### 3.2. Blockchain Technologies as a Means of Secure Data Storage

The security issues faced in the healthcare sector make patients rightfully unsure of the security of their medical information. However, a recent technological innovation in data storage/transfer may prove to be viable in mitigating breaches due to external attacks. Blockchain technology was most notably pioneered by the cryptocurrency “Bitcoin”. It was developed as a means to solve the double-spending problem, in which a sender makes two transactions of the same currency simultaneously in a decentralized network, which lacks the standard central medium that would validate or invalidate transactions and decide transaction order [62]. The basic premise involves broadcasting transaction information, privatized through a private and public key-pair system, to all nodes with the network. Each node then competes to solve a cryptographic problem, locating a value that, when hashed, produces a hash with a predetermined number of 0 bits. When a node is successful, it broadcasts that success to all other nodes in the network. If the nodes deem the transaction valid, the newly created block is linked to the preceding block, and nodes begin working on successive blocks, adding on to the chain. [62]. Blocks are linked to each other by their headers, which contain hashes of the previous block’s header as well as a hash of the transactions; this ensures continual transaction validity, as attempting to change a block’s transaction information will change the value of the hash and break the chain [63]. A general summary of this process is depicted in Figure 2.

Notably, nodes in a blockchain network will preferentially build off of the current longest-running chain. This property makes blockchains nearly immutable by attackers, as modifying the information contained within a blockchain requires an attacker to create an entirely new chain, starting with the target, that is at least one block longer than the original; the likelihood of this being possible decreases with the overall computational power of the network’s userbase, as an attacker would need to replace all of the nodes following the target node in order to successfully overwrite the target; this is extremely unlikely if the attacker does not have overpowering computational resources [62]. There is a possibility for a so-called “51% attack”, in which an attacker is able to control over 50% of the network’s computational power. However, this is not an issue for private blockchain networks where the introduction of large numbers of malicious nodes is infeasible. The overall decentralization of the blockchain system has an advantage over those that rely on a central entity, as the network does not have a single point of failure and can thus continue to function even if some nodes are inoperable [64].

Though the basic blockchain implementation is exceptional at preventing invalid modifications, there are problems that are impermissible in a healthcare environment. An inherent problem in applying such methods to medical data is that transactions are publicly visible to all nodes on the network. Additionally, while steps are taken to ensure user privacy through the private-public key authorization system, it is possible to identify users if enough data identified to one public key can be aggregated [65].

### 3.3. Potential Applications of Blockchain Technologies in the Healthcare Sector

Relatively recent implementations of the blockchain methodology have yielded solutions to many of these problems. Many require the authorization of nodes, limiting the processing of blocks only to properly permitted nodes. The so-called “smart” blockchain platforms, which include Ethereum, NEO, and NEM utilize both public and private chains; they retain many of the benefits of fully public blockchains through access verification and control, while limiting access to private or sensitive information [66]. Immutability of data is also a pitfall of the traditional blockchain structure, as individuals may want the ability to destroy their own data in certain circumstances; private blockchain implementations, such as Enterprise, can allow for the deletion of data [67]. Data can also be encrypted by assigning patients a private key for use as a decryption key; such a mechanism would render data unreadable to any infiltrating entity [68]. In addition, existing systems such as FHIRChain utilize public blockchains that hold pointers to privately stored data, rather than the data itself [69]. Furthermore, they require multi-signature contracts in which parties other than the sender and the recipient must authorize the transaction. Such an implementation could be used to require patient authorization for a transfer of medical records or health information between institutions [66]. Much of the allure of blockchain-based healthcare systems is in the heightened ability of the patient to control his or her data, including them in the process of data sharing and ensuring that confidential information regarding the patient is not disseminated without their consent. Along with this comes an unprecedented ability to share and verify records across institutions easily without involving a third party, providing a great boon to research efforts [69]. Blockchain approaches have been devised for pharmacogenomic testing [70], and some consumer DNA sequencing services have employed consumer-controlled blockchain systems where individuals can be sequenced; these systems allow users to have access to their own data, and users have the ability monetize and grant access to their own genetic information. Such implementations help to counter concerns over what sequencing companies can do with their data [67]. Similar approaches may apply to the healthcare sector, where concerns over genetic data sharing, and the ability to give and revoke data-sharing consent, are relevant [71].

The security provided by blockchain technologies may extend beyond simple EHR storage as well. Given the difficulty of diagnosing AD, some have suggested that wearable technologies and other internet-of-things (IOT) enabled devices may prove to be valuable in the diagnosis of AD. Deviations in sleep, navigation, driving ability speech, and a host of other behaviors have been found to be related to AD and early-stage cognitive impairment [72,73]. These behaviors can be difficult to measure in a clinical setting, and may suffer from daily variations (i.e., a patient’s “good days” and “bad days”) smartphone/smartwatch applications, and other IOT enabled sensors and devices can collect vast quantities of data on these so-called “digital biomarkers” for analysis, ideally making them more precise as diagnostic tools [72,73]. The use of these technologies as diagnostic tools for neurodegenerative disorders is still in its infancy, and studies of their efficacy are fairly limited [74]. 

While there are many challenges that need to be addressed before the widespread deployment of such technologies, one particular element of concern is patient privacy and trust [72,73,75]. Indeed, information privacy, the overall ability of an individual to control their own information, is a considerable concern with regard to health-related assistive technology in general [76]. An analysis of electronic informed consent for the mPower Parkinson’s disease application mediated research process found that individuals with Parkinson’s disease were almost three times more likely to have data-privacy related concerns than controls. Furthermore, participants in general appreciated having access to their own data, with researchers noting that participant complaints arose from any difficulties or deficiencies in data access [77]. Blockchain-based systems may offer the patient data control, access, and security needed to be comfortable using such applications and IOT connected devices [67,78,79].

Smart blockchain technologies that include privacy-centric features such as data encryption and/or private chains for the actual data, may well be the next step in secure information storage. We propose that widespread adoption of blockchain based medical data systems, may improve patients’ confidence that their information and health records will remain confidential. The heightened security can dissuade and mitigate attacks, while simultaneously allowing for patient-centric information privacy and sharing. This may inspire confidence in the information-security of the healthcare sector, and help patients feel more comfortable sharing concerns about their health and cognitive wellbeing with their physicians, allowing physicians to keep them informed and perform tests as necessary. Though this does not directly address institutional stigma, which primarily must be solved through legislation, it will limit the number of unauthorized and unknown institutions possessing patients’ data, lessening their fears of institutional stigma. A summary of this proposal can be found in Figure 3.

## 4. Conclusions

AD stigma poses a rather complicated problem for AD diagnosis, treatment, and prevention. The nature of a currently uncurable and unpreventable disease that effects one’s cognitive capacity leads to considerable stigma as individuals are associated with symptoms they may or may not have. This causes people to avoid diagnosis and the social and institutional problems it entails. Much of this stigma derives from misinformation about the disease and negative opinions that are spread online. The use of NLP to assess and monitor the sentiment of the public may be invaluable in both quantifying the prevalence of stigma, and in analyzing the success of anti-stigma campaigns, as little is known about the actual effects of efforts made to combat stigma. As information security is of considerable concern to the general public, even more so to those with sensitive health information they would rather keep private, enhanced information security such as that provided with decentralized databases like a blockchain based service may prove invaluable in fighting AD stigma. Fighting AD stigma is an essential part of maximizing quality of life and treatment options for both current and future AD patients.

## Figures and Tables

**Figure 1 brainsci-10-00183-f001:**
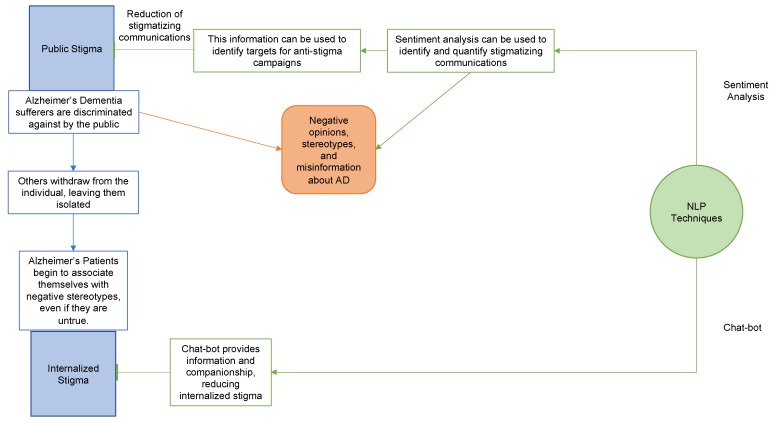
Summary of proposed NLP applications. We propose that an NLP-based sentiment analysis could be used to identify the need for and facilitate the deployment of anti-stigma measures aimed at minimizing public stigma towards Alzheimer’s dementia and other dementia patients. A chat-bot/AI companion implementation would aid in the independent living of AD patients while alleviating social isolation and the feelings of loneliness that exacerbate self-stigma.

**Figure 2 brainsci-10-00183-f002:**
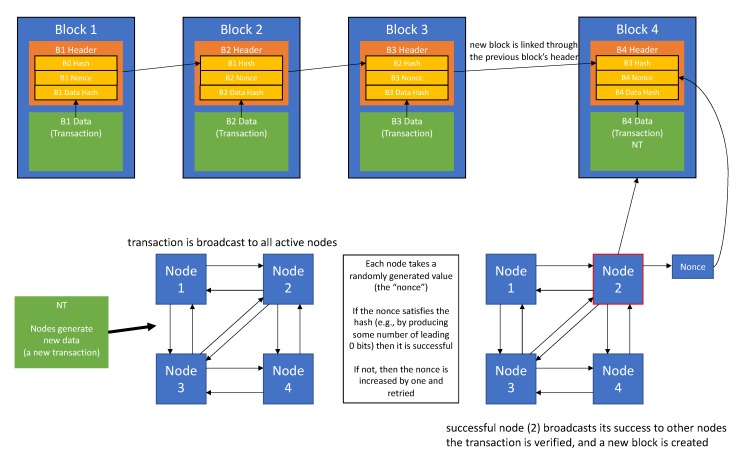
Summary of basic blockchain methodology. When transactions are made that need to be stored in the chain. nodes compete to find a “nonce” that satisfies the hash-problem assigned to them. If the data is valid, the successful node creates a block, which is chained to the most recently created block in the longest chain.

**Figure 3 brainsci-10-00183-f003:**
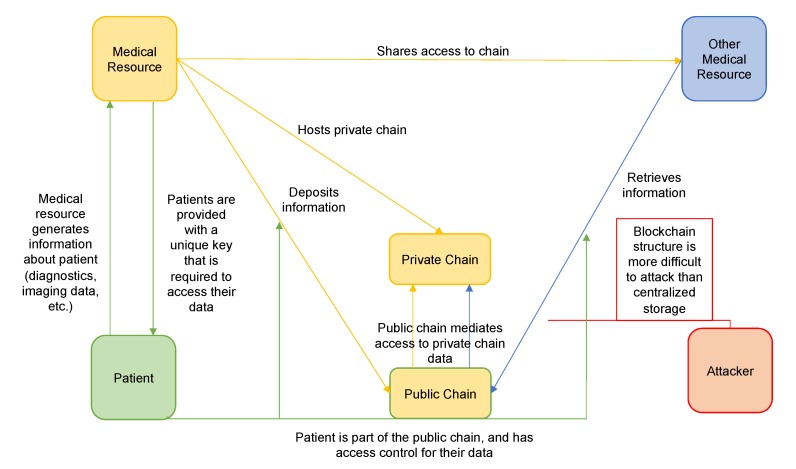
Summary of Blockchain Proposal. A successful blockchain implementation of an EHR system could provide a more secure and patient-privacy-oriented network, that gives patients confidence in the confidentiality of their information and diagnosis.

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
