# Peer review of "Overcoming Alzheimer’s Disease Stigma by Leveraging Artificial Intelligence and Blockchain Technologies"

_brainsci, 2020, doi:10.3390/brainsci10030183_

Round 1
Reviewer 1 Report
This paper review how recent advents in information technology may be employed to help fight this stigma. The paper is well structured and organized. While figures are attractive and well designed, they should improve the flow of concepts, which is not clear in the current MS. For example, in Fig 1, the starting-ending cycle priorities are not clear, and the logic in Figure 3 is lost.
Also, authors should consider mentioning the challenges of blockchain in healthcare that may also impact stigma, especially the one associated with the immutability of data (i.e., once data are entered, they cannot be removed). For example, provided, we can find the cure for AD and assuming a patient may be cured, how this outcome will impact the patient's future information and associated stigma. What if we can't find the AD cure, but the patient changes his/her lifestyle conditions and improves his/her cognition function. How could this change be reflected in the blockchain-based EHRs and unlinked to potential-stigma?
Author Response
Reviewer 1’s comments and responses
This paper review how recent advents in information technology may be employed to help fight this stigma. The paper is well structured and organized. While figures are attractive and well designed, they should improve the flow of concepts, which is not clear in the current MS. For example, in Fig 1, the starting-ending cycle priorities are not clear, and the logic in Figure 3 is lost.
Response: We have made an effort to simplify figures 1 and 3.
Also, authors should consider mentioning the challenges of blockchain in healthcare that may also impact stigma, especially the one associated with the immutability of data (i.e., once data are entered, they cannot be removed). For example, provided, we can find the cure for AD and assuming a patient may be cured, how this outcome will impact the patient's future information and associated stigma. What if we can't find the AD cure, but the patient changes his/her lifestyle conditions and improves his/her cognition function. How could this change be reflected in the blockchain-based EHRs and unlinked to potential-stigma?
Response: In the example cases you mentioned, however, the fact the individual had AD is still medically relevant. Ideally, if the patient can be confident in the security of that information, it should not be a concern that it exists. In the event the patient wants that data deleted, however, implementations of private blockchains such as “Enterprise” allow for that possibility, and we have made note of this in the text (line 335).
Reviewer 2 Report
Pilozzi and Huang have an interesting analysis of the use of natural language processing to protect Alzheimer's patients from stigma and also protect them from social isolation. However, the blockchain section is a very simple synopsis of the field with little to no critical analysis. This section should be removed, and the article should be focused on the natural language processing/Alzheimer's theme before consideration for publication. All points in their discussion of blockchain can be found in the article International Journal of Medical Informatics, 134, (2020) 104040. Other articles exist as well which provide a detailed analysis of blockchain in healthcare informatics.
Minor:
Modify “AD is unique in the societal challenges it poses.” to “Dementia, the most prevalent form of which is caused by Alzheimer’s disease, is unique in the societal challenges it poses.”
There are many kinds of prevalent dementia like Lewy Body, vascular, Parkinson’s etc., and the sentences following this sentence in the manuscript fail to show the unique challenges posed by AD. If the authors would protest that the level of stigma that Alzheimer’s faces is higher, they should investigate how stigmatizing and isolating frontotemporal dementia, Parkinson’s-related dementia and vascular dementia are.
Modify “The diagnosis of carries an association…” to “The diagnosis of AD carries an association…”
Additional superfluous commas and spacing issues need attention in this manuscript.
Modify “AD is unique in its situation as a physical neurodegenerative ailment with cognitive effects, with aspects of both physical debilitation and mental illness stigma that blend together”
This is not true. This is actually a feature of dementia not just Alzheimer’s disease. Lewy body dementia, Huntington’s disease and frontotemporal dementia all fit this “situation.”
Modify “Furthermore, there are notable differences in the perception of and aging between cultures.”
Perception of what? Should the "and" be removed?
Modify “While in an ideal world, people would feel comfortable and able to learn of their status and freely discuss it with others, that is not necessarily the case.”
"...learn of their Alzheimer’s disease status and..."
Major:
Citation Needed for “Based on manual classification, around 40% of health/healthcare-related tweets exhibited some form of sentiment.”
The details of this statistic should also briefly be discussed in a sentence.
The paragraph beginning on line 137 of page 4 needs to be reworked.
This paragraph, which discusses the use of human agents, is in between two paragraphs discussing the use of robotic agents, and it should be discussed thoroughly whether the research conclusions of the human-controlled “wizard of oz” study would be applicable if the human were to be replaced with an AI, especially given the findings of study [40] presented in the next paragraph. Otherwise, as it currently stands, there is a massive conflict in the presented research left implicit and unaddressed by the authors. An excellent research question emerges from this paragraph is “given the siccess of human-controlled avatars, could we successfully replace the human with an AI when it is known that the elderly have apprehension with regard to the idea of a robot?” This needs to be discussed.
Citation Needed for “While protections exist to mitigate genetic discrimination by employers and insurance companies, there are no comparable protections for biomarker tests or imaging.”
Blockchain section. Very little of this is new insight and most points of this manuscript are made in International Journal of Medical Informatics, 134, (2020) 104040. The Blockchain section should be removed and the AI/NLP section should be expanded before publication.
Author Response
Reviewer 2’s comments and responses
Pilozzi and Huang have an interesting analysis of the use of natural language processing to protect Alzheimer's patients from stigma and also protect them from social isolation. However, the blockchain section is a very simple synopsis of the field with little to no critical analysis. This section should be removed, and the article should be focused on the natural language processing/Alzheimer's theme before consideration for publication. All points in their discussion of blockchain can be found in the article International Journal of Medical Informatics, 134, (2020) 104040. Other articles exist as well which provide a detailed analysis of blockchain in healthcare informatics.
Minor:
Modify “AD is unique in the societal challenges it poses.” to “Dementia, the most prevalent form of which is caused by Alzheimer’s disease, is unique in the societal challenges it poses.”
There are many kinds of prevalent dementia like Lewy Body, vascular, Parkinson’s etc., and the sentences following this sentence in the manuscript fail to show the unique challenges posed by AD. If the authors would protest that the level of stigma that Alzheimer’s faces is higher, they should investigate how stigmatizing and isolating frontotemporal dementia, Parkinson’s-related dementia and vascular dementia are.
Response: We have changed this and other parts of the paper to refer to dementia more generally where appropriate, and we have added a note that Alzheimer’s disease is the most common cause of dementia (line 27).
Modify “The diagnosis of carries an association…” to “The diagnosis of AD carries an association…”
Response: We have made the requested change, thank you for pointing this out (line 36).
Additional superfluous commas and spacing issues need attention in this manuscript.
Response: We have made a pass through the paper to find and remove extra commas and spaces.
Modify “AD is unique in its situation as a physical neurodegenerative ailment with cognitive effects, with aspects of both physical debilitation and mental illness stigma that blend together”
This is not true. This is actually a feature of dementia not just Alzheimer’s disease. Lewy body dementia, Huntington’s disease and frontotemporal dementia all fit this “situation.”
Response: You are correct. We have altered the paper to include dementia more generally where applicable and note that Alzheimer’s disease is the most common cause of dementia.
Modify “Furthermore, there are notable differences in the perception of and aging between cultures.”
Perception of what? Should the "and" be removed?
Response: This should say “… of dementia and aging between cultures”. We have corrected this, thank you for pointing this out. (line 227)
Modify “While in an ideal world, people would feel comfortable and able to learn of their status and freely discuss it with others, that is not necessarily the case.”
"...learn of their Alzheimer’s disease status and..."
Response: We have altered the sentence as requested; thank you for pointing this out. (line 259)
Major:
Citation Needed for “Based on manual classification, around 40% of health/healthcare-related tweets exhibited some form of sentiment.”
The details of this statistic should also briefly be discussed in a sentence.
Response: The statement has been changed somewhat to correspond to a different reference due to flaws in the original paper. More details have been added as well (line 113).
The paragraph beginning on line 137 of page 4 needs to be reworked.
This paragraph, which discusses the use of human agents, is in between two paragraphs discussing the use of robotic agents, and it should be discussed thoroughly whether the research conclusions of the human-controlled “wizard of oz” study would be applicable if the human were to be replaced with an AI, especially given the findings of study [40] presented in the next paragraph. Otherwise, as it currently stands, there is a massive conflict in the presented research left implicit and unaddressed by the authors. An excellent research question emerges from this paragraph is “given the siccess of human-controlled avatars, could we successfully replace the human with an AI when it is known that the elderly have apprehension with regard to the idea of a robot?” This needs to be discussed.
Response: As requested, we have significantly expanded the section on AI companions, addressing the issues of acceptance among the elderly/dementia patients as well.
Citation Needed for “While protections exist to mitigate genetic discrimination by employers and insurance companies, there are no comparable protections for biomarker tests or imaging.”
Response: We have added the citation (line 239).
Blockchain section. Very little of this is new insight and most points of this manuscript are made in International Journal of Medical Informatics, 134, (2020) 104040. The Blockchain section should be removed and the AI/NLP section should be expanded before publication.
Response: We have not removed this section as we feel it is an integral part of this review. However, we have expanded it to include more specific relevance to AD-related concerns, rather than focusing on only general EHR security.
Round 2
Reviewer 2 Report
Significant improvements have been made to this manuscript and the blockchain section now adds important points of consideration to the management of AD. All of my previous points have been addressed.